*Review Article*

# Whole-sporozoite malaria vaccines: where we are, where we are going

Diana Moita & Miguel Prudêncio 🄳 ✉

## Abstract

The malaria vaccination landscape has seen significant advancements with the recent endorsement of RTS,S/AS01 and R21/Matrix-M vaccines, which target the pre-erythrocytic stages of *Plasmodium falciparum* (Pf) infection. However, several challenges remain to be addressed, including the incomplete protection afforded by these vaccines, their dependence on a single Pf antigen, and the fact that they were not designed to protect against *P. vivax* (Pv) malaria. Injectable formulations of whole-sporozoite (WSpz) malaria vaccines offer a promising alternative to existing subunit vaccines, with recent developments including genetically engineered parasites and optimized administration regimens. Clinical evaluations demonstrate varying efficacy, influenced by factors, such as immune status, prior exposure to malaria, and age. Despite significant progress, a few hurdles persist in vaccine production, deployment, and efficacy in malaria-endemic regions, particularly in children. Concurrently, transgenic parasites expressing Pv antigens emerge as potential solutions for PvWSpz vaccine development. Ongoing clinical studies and advancements in vaccine technology, including the recently described PfSPZ-LARC2 candidate, signify a hopeful future for WSpz malaria vaccines, which hold great promise in the global fight against malaria.

**Keywords** *Plasmodium falciparum*; *Plasmodium vivax*; Clinical Evaluation; PfSPZ; Vaccination Regimens
**Subject Category** Microbiology, Virology & Host Pathogen Interaction

## Introduction

After decades of research and dozens of candidates in different stages of pre-clinical or clinical development, the malaria vaccination field recently took a major leap forward, with the endorsement of two vaccines by the World Health Organization (WHO). In 2021, RTS,S/AS01 (hereafter referred to as RTS,S) became the first-ever vaccine to receive approval for use in children at risk of malaria (D'Souza and Nderitu, 2021). Although some argued that the WHO's recommendation might have been too hasty (Bjorkman et al, 2023), the vaccine has since been implemented through national immunization programs in Ghana, Kenya and Malawi, leading to substantial reductions in hospital admissions with severe malaria (Asante et al, 2024). Two years after its endorsement, RTS,S was joined by a second vaccine, termed R21/Matrix-M (hereafter R21) (Datoo et al, 2024; Moorthy and Binka, 2021). While some reports indicate that R21 may be more effective than RTS,S, the two vaccines have not yet been evaluated in a direct, head-to-head manner, and there is no evidence to date showing one vaccine performs better than the other (WHO, 2024). Like with any vaccine, the safety and efficacy of both RTS,S and R21 will be continuously monitored, and their public health benefits across sub-Saharan Africa quantified (Schmit et al, 2024).

RTS,S and R21 are subunit vaccines that target the pre-erythrocytic stages of infection by *Plasmodium falciparum* (Pf), the deadliest human-infective malaria parasite. Both employ a fusion of the immunodominant epitopes of the Pf circumsporozoite protein (PfCSP) with a surface antigen of the hepatitis B virus, and are formulated with the AS01 and Matrix-M adjuvants, respectively, to maximize antibody responses (Datoo et al, 2024; Laurens, 2020). However, in the absence of complete sterile immunity, and since immune responses induced by RTS,S and R21 do not target gametocytes, transmission is not fully abrogated by these vaccines (Zavala, 2022). Work by Read et al, indicating that the deployment of partially effective vaccines may increase overall pathogen virulence (Read et al, 2015), is also highly relevant in this regard. Additionally, concerns about the possible spread of parasite strains not targeted by either of these vaccines have arisen from reports of selective pressure with reduced protection against infections with RTS,S-mismatched PfCSP alleles (Neafsey et al, 2015). Lastly, neither RTS,S nor R21 have been designed to target *P. vivax* (Pv) the most geographically widespread human malaria parasite, leaving this hurdle to malaria elimination largely unaddressed.

Whole-sporozoite (WSpz) malaria vaccines rely on the administration of *Plasmodium* sporozoites, the liver-infective forms of malaria parasites, as immunization agents. To prevent the development of the symptomatic blood stage of infection following vaccine inoculation, parasites may be attenuated by irradiation (radiation-attenuated sporozoites, RAS) or genetic modification (genetically-attenuated parasites, GAP), or be administered under the prophylactic cover of a drug such as chloroquine, which specifically targets the parasite's erythrocytic stages (chemoprophylaxis and sporozoites, CPS) (reviewed in (Mendes et al, 2017)). More recently, this list has been expanded to include genetically engineered rodent *P. berghei* (Pb) parasites expressing antigens of

Instituto de Medicina Molecular João Lobo Antunes, Faculdade de Medicina, Universidade de Lisboa, Av. Prof. Egas Moniz, 1649-028 Lisboa, Portugal.
✉E-mail: mprudencio@medicina.ulisboa.pt

**Glossary**

| | | | |
|---|---|---|---|
| Subunit vaccine | a type of vaccine that contains only specific components or antigens of a pathogen, rather than the entire organism, to elicit a targeted immune response in the vaccine's recipient. | | of sporozoites or parasitized red blood cells. |
| | | Homologous challenge | controlled human malaria infection employing the same parasite strain as the one used in the vaccine. |
| Whole-sporozoite vaccine | a type of vaccine that employs sporozoites, the liver-infective forms of the malaria parasite, either live or attenuated, in order to elicit immune protection against the pre-erythrocytic stages of the parasite. | Heterologous challenge | controlled human malaria infection employing a parasite strain different from one used in the vaccine. |
| | | Immune tolerance | immune system's state of unresponsiveness to an antigen that would otherwise trigger an immune response. |
| Cryopreservation | the process of preserving cells, tissues, or organisms at extremely low temperatures to maintain their viability. | Immunomodulation | alteration of the immune system that can either enhance or suppress the immune response to a stimulus. |
| Good manufacturing practices | a system that ensures that medicinal products are consistently produced and controlled according to quality standards appropriate to their intended use and as required by the product specification. | Cost of goods | total cost incurred to produce a product, which, in the case of a vaccine, includes its production and delivery. |
| Controlled human malaria infection | deliberate infection with malaria parasites either by mosquito bite or by direct injection | T-cell epitope | a specific region of an antigen that is recognized by T cells and elicits an immune response. |

their human-infective counterparts (PbVac) as platforms for immunization against human malaria (Mendes et al, 2018a; Moita et al, 2022a).

In this short review, we address current challenges and future directions for WSpz malaria vaccination, and discuss its prospective role in the global fight against malaria.

## Clinical evaluation of WSpz malaria vaccines

The potential for successful sporozoite-based vaccination against malaria was initially established in 1967 by Ruth Nussenzweig's demonstration that intravenous inoculation of rodent PbRAS into mice conferred protections against murine malaria (Nussenzweig et al, 1967). Since sporozoites obtained by dissection of mosquito salivary glands contain extraneous debris that could pose medical risks, testing this immunization strategy in humans required that infected mosquitoes be X-irradiated and then allowed to feed on volunteers (Vanderberg, 2009). Thus, only six years after publication of the Nussenzweig study, protection of humans against malaria through inoculation of PfRAS delivered by mosquito bite was demonstrated for the first time (Clyde et al, 1973). This remarkable observation opened up the attractive possibility that *Plasmodium* sporozoites could indeed be used for mass vaccination against malaria. However, many thought that such an ambitious goal would be unattainable, such was the number and the magnitude of the hurdles that would need to be overcome before it could even begin to be contemplated (Druilhe and Barnwell, 2007; Luke and Hoffman, 2003). For the most part, these challenges resulted from the fact that the administration of such vaccines to humans was entirely dependent on their delivery by mosquito bite, which not only posed major ethical concerns but also raised important questions regarding production and efficacy. Overcoming these concerns demanded that sufficient numbers of sporozoites that complied with regulatory, potency and safety requirements could be produced and administered by a clinically practical route (Luke and Hoffman, 2003). Add to this the success, popularity, and ease-of-production of subunit

vaccination approaches (Plotkin, 2014), and it is unsurprising that only a few clinical trials of WSpz vaccines were conducted for over two-and-a-half decades since the first-in-humans PfRAS immunization, leaving the field at what appeared to be a dead end (Druilhe and Barnwell, 2007).

In spite of these seemingly insurmountable challenges, in 2003, a path to developing a non-replicating, metabolically active, radiation-attenuated Pf sporozoite vaccine was outlined. This included a structured plan to develop and optimize the production of cryopreserved, adequately radiation-attenuated, aseptic, purified, Pf sporozoites suitable for administration by a clinically practical route (Luke and Hoffman, 2003). These efforts culminated in the manufacture of the Good Manufacturing Practices (GMP)-compliant PfSPZ vaccine that met regulatory requirements for administration to humans (Hoffman et al, 2010; James et al, 2022).

The demonstration that PfSPZ vaccine efficacy depended on its administration by intravenous, but not intradermal or subcutaneous injection (Epstein et al, 2011; Seder et al, 2013), opened up a new era for the field of WSpz malaria vaccination. This led to the multiplication of clinical trials employing injectable vaccine formulations, as the number of human studies resorting to vaccine administration by mosquito bite dwindled (Nunes-Cabaco et al, 2022). We recently reviewed the results of the 32 clinical trials of WSpz vaccines published until late 2021, focusing on the pre-clinical development and clinical history of the various candidates, and on the assessment of their immunogenicity in humans (Nunes-Cabaco et al, 2022). We now provide an updated version of this compilation of clinical studies, in a format that is both readily accessible and easy to navigate. The printed version of this repository (Table 1) is accompanied by its digital counterpart, found at https://miguelprudencio.com/vaccine-map/, which offers search, filter and data export features that facilitate access to information organized by vaccine type, phase of clinical development, region where the trial was conducted, age group, status of previous exposure to malaria, immunization route and vaccination regimen, type and time of (re-)challenge, and protective efficacy observed.

**Table 1.  Repository of whole-sporozoite malaria vaccine clinical trials.**

| Type of WSp vaccine | Clinical trial phase | NCT | Study population: Age group | Malaria-naïve or -exposed subjects | Nr. of subjects | Immunization: Administration route | Dose | Number of doses | Schedule | Antimalarial drug | Challenge: Type of challenge | Administration route | Dose | Homologous/ Heterologous | Time-point | Nr. protected subjects/Total | Rechallenge: Information | Nr. protected subjects/Total | DOI |
|---|---|---|---|---|---|---|---|---|---|---|---|---|---|---|---|---|---|---|
| | | | | | | | | | | | | | | *X-irradiated mosquitoes* | | | | |
| | N/A | N/A | Adults (M; 24-34yo) | Malaria-naïve | 3 | Mosquito bite | 379 bites (total) from Pf-infected mosquitoes | 6 | Irregular intervals over 84d | N/A | CHMI | Mosquito bite | 9-13 mosquitoes | Homologous | 14d after the last immunization | 1/3 | The protected subject received 5 more immunizations (819 mosquitoes total). The subject remained protected for 2mo against rechallenge. | 1/1 | 10.1097/00000441-197309000-00002 |
| | N/A | N/A | Adult (M; 37yo) | Malaria-naïve | 1 | Mosquito bite | 1441 bites (total) from Pf-infected mosquitoes | 13 | Irregular intervals over 420d | N/A | CHMI | Mosquito bite | 9 mosquitoes | Homologous | Day 98 | 1/1 | The subjected underwent the following rechallenges: a) Day 327; Homologous; 13 mosquitoes b) Day 413; Heterologous (Malaya strain); 14 mosquitoes c) Day 435; P. vivax; 14 mosquitoes d) Day 448; Heterologous (Panama II strain); 6 mosquitoes e) Day 459; Heterologous (Philippines strain); 11 mosquitoes | a) 1/1 b) 1/1 c) 0/1 d) 1/1 e) 1/1 | 10.1097/00000441-197312000-00001] |
| | N/A | N/A | Adult (M; 48yo) | Malaria-naïve | 1 | Mosquito bite | 838 bites (total) from Pf-infected mosquitoes | 2 | Irregular intervals between days -205 and -84 | N/A | CHMI | Mosquito bite | 8 mosquitoes | Homologous | Day -77 (1wk after the last immunization) | 0/1 | a) The subject received further immunizations (968 mosquitoes total) between -55d and -1d and was rechallenged with Pf-infected mosquitoes on day 0 b) The subject underwent a 2nd challenge with Pf-infected mosquitoes 24d after the previous one | a) 1/1 b) 1/1 c) 1/1 d) 1/1 e) 0/1 | 10.1038/nbt0997-876 10.1097/00000441-197309000-00002 |
| | | | | | | | | | | | | | Pv-infected mosquitoes | | 0/1 | The subjected underwent the following immunization sessions: with irradiated Pv-infected mosquitoes (180 total) between 31d and 43d; with irradiated Pf-infected mosquitoes between 118d and 135d (400 total) and with irradiated Pv-infected mosquitoes (359 total) between 136d and 148d. The subject subsequently underwent 3 rechallenges with Pv-infected mosquitoes on c) 161d (11 mosquitoes); d) 237d (12 mosquitoes) and e) 330d (9 mosquitoes) | | |
| | N/A | N/A | Adults (M; 21-40yo) | Malaria-naïve | 4 | Mosquito bite | < 200 bites (total) from Pf-infected mosquitoes | 2 - 4 | Irregular intervals over 1-4mo | N/A | CHMI | Mosquito bite | 13-14 mosquitoes | Homologous | 2wk after the last immunization | 0/4 | N/A | N/A | N/A |
| | | | | | 2 | | 440-987 bites (total) from Pf-infected mosquitoes | 6 - 8 | Irregular intervals over 10-38wk | | | | | Homologous | 2wk after the last immunization | 2/2 | a) 1 of the protected subjects was homologously rechallenged 16wk after the last immunization b) The other protected subject underwent homologous, heterologous, and homologous rechallenges 8, 17 and 25wk, respectively, after the last immunization. The subject was protected following the 1st rechallenge but became infected after the subsequent one | a) 0/1 b) 0/1 | |
| | | | | | 1 | | | | | | | | | Heterologous | 8wk after the last immunization | 1/1 | The protected subject was homologously rechallenged 18wk after the last immunization | 0/1 (0%) | |
| | N/A | N/A | Adults (M; 18-50yo) | Malaria-naïve | 11 | Mosquito bite | 1001-2927 bites (total) from Pf-infected mosquitoes | 9 (avg) | Biweekly over 9-10mo | N/A | CHMI | Mosquito bite | 5 mosquitoes | Homologous | 2-9wk after the last immunization | 10/11 | 5 protected subjects were rechallenged 23-42wk; 2 subjects were heterologously rechallenged (Pf 7G8 strain) | 4/5 2/2 | 10.1086/339409 |
| | N/A | N/A | Adults (M; 23-33yo) | Malaria-naïve | 2 | Mosquito bite | 625 and 715 bites (total) from Pf-infected mosquitoes | 7 and 11 | Irregular intervals over 93 and 276d | N/A | CHMI | Mosquito bite | 5 mosquitoes | Homologous | 36d after the last immunization | 0/2 | N/A | N/A | 10.1093/infdis/168.4.1066 |
| | | | | | 3 | | 1563-1681 bites (total) from Pf-infected mosquitoes | 19 | Irregular intervals over 88d | | | | | | 25d after the last immunization | 3/3 | 1 of the protected subjects received a series of booster immunizations ~3mon after the 1st challenge and was rechallenged 9mo after that. | 1/1 | 10.4269/ajtmh.1991.45.539 |
| | 2 | NCT01082341 | Adults (M & F; 18-45yo) | Malaria-naïve | 12 | Mosquito bite | ~434 bites (total) from Pv-infected mosquitoes | 7 | 9wk intervals | N/A | CHMI | Mosquito bite | 2-4 mosquitoes | Homologous | 8wk after the last immunization | 5/12 | N/A | N/A | 10.1371/journal.pntd.0005070 |
| | | | | | | | | | | | | | | *Sanaria® PfSPZ Vaccine* | | | | |
| | 1 | NCT01001650 | Adults (M & F; 18-50yo) | Malaria-naïve | 11 | id or sc | 7.5 x 10³ spz | 4 | 4wk intervals | N/A | CHMI | Mosquito bite | 5 mosquitoes | Homologous | 3wk after the last immunization | 0/11 | N/A | N/A | 10.1126/science.1211548 |
| | | | | | 16 | | 3 x 10⁴ spz | 4 | 4wk intervals | | | | | | | 2/16 | N/A | N/A | |
| | | | | | 17 | | 1.35 x 10⁵ spz | 6 | 4wk intervals and 12wk interval between doses 5 and 6 | | | | | | | 0/17 | N/A | N/A | |
| | 1 | NCT01441167 | Adults (M & F; 18-45yo) | Malaria-naïve | 3 | iv | 7.5 x 10³ spz | 4 | 4wk intervals | N/A | CHMI | Mosquito bite | 5 mosquitoes | Homologous | 3wk after the last immunization | 0/3 | N/A | N/A | 10.1126/science.1241800 10.1038/nm.4110 |
| | | | | | 3 | | | 6 | 4wk interval and 8wk interval between doses 2 and 3 | | | | | | | 0/3 | | | |
| | | | | | 9 | | 3.0 x 10⁴ spz | 4 | 4wk intervals | | | | | | | 1/9 | | | |
| | | | | | 2 | | | 6 | | | | | | | | 0/2 | | | |
| | | | | | 9 | | 1.35 x 10⁵ spz | 4 | | | | | | | | 6/9 | 6 protected subjects from these groups underwent repeat CHMI 21wk after the last immunization. | 2/6 | |
| | | | | | 6 | | | 5 | | | | | | | | 6/6 | | | |
| | 1 | NCT01441167 | Adults (M & F; 18-45yo) | Malaria-naïve | 9 | iv | 2.7 x 10⁵ spz | 3 | Wk 0, 4 and 20 | N/A | CHMI | Mosquito bite | 5 mosquitoes | Homologous | 3wk after the last immunization | 3/9 | The 3 protected subjects underwent rechallenge 21-25wk after the last immunization | 2/3 | 10.1038/nm.4110 |
| | | | | | 9 | | | 4 | Wk 0, 4, 8 and 20 | | | | | | | 7/9 | Four protected subjects underwent rechallenge a) 21-25wk after the last immunization and one of the protected subjects followinf the 1st rechallenge underwent another challenge b) 59wk after the last | a) 3/4 b) 1/1 | |
| | | | | | 11 | | | 4 | | | | | | | 21-25wk after the last immunization | 6/11 | Four protected subjects underwent rechallenge 59wk after the last immunization | 4/4 | |
| | | | | | 12 | | 1.35 x 10⁵ + 4.5 x 10⁵ spz | 4 + 1 | Wk 0, 4, 8, 12 and 20 | | | | | | 3wk after the last immunization | 8/12 | Four protected subjects underwent rechallenge 21-25wk after the last immunization | 4/7 | |
| | | | | | 8 | im | 2.2 x 10⁶ spz | 4 | Wk 0, 4, 8 and 20 | | | | | | | 3/8 | Three protected subjects underwent rechallenge 21-25wk after the last immunization | 0/3 | |
| RAS | 1 | NCT02215707 | Adults (M & F; 18-45yo) | Malaria-naïve | 13 | iv | 2.7 x 10⁵ spz | 5 | Wk 0, 4, 8, 12 and 20 | N/A | CHMI | Mosquito bite | 5 mosquitoes | Homologous | 3wk after the last immunization | 12/13 | N/A | N/A | 10.1172/jci.insight.89154 |
| | | | | | 5 | | | 5 | | | | | | Heterologous | 3wk after the last immunization | 4/5 | | | |
| | | | | | 10 | | | 5 | | | | | | Homologous | 24wk after the last immunization | 7/10 | | | |
| | | | | | 10 | | | 5 | | | | | | Heterologous | 24wk after the last immunization | 1/10 | | | |
| | | | | | 15 | | 4.5 x 10⁵ spz | 3 | Wk 0, 8 and 16 | | | | | Homologous | 3wk after the last immunization | 13/15 | | | |
| | | | | | 14 | | | 3 | | | | | | Homologous | 24wk after the last immunization | 8/14 | | | |
| | 1 | NCT02015091 | Adults (M & F; 20-40yo) | Malaria-naïve | 14 | iv | 9.0 x 10⁵ spz | 3 | 8wk intervals | N/A | CHMI | Mosquito bite | 5 mosquitoes | Homologous | 19wk after the last immunization | 9/14 | 6 protected subjects underwent repeat heterologous (Pf 7G8 strain) CHMI 33wk after the last immunization | 5/6 | 10.1073/pnas.1615324114 |
| | 1 | NCT01988636 | Adults (M & F; 18-35yo) | Malaria-exposed (Mali) | 41 | iv | 2.7 x 10⁵ spz | 5 | Days 0, 28, 56, 84 and 140 | N/A | Natural transmission | N/A | N/A | N/A | 24wk surveillance post-vaccination | 14/41 | N/A | N/A | 10.1016/S1473-3099(17)30104-4 |
| | 1 | NCT02613520 | Adults (M & F; 18-45yo) | Malaria-exposed (Tanzania) | 5 | iv | 9.0 x 10⁵ spz | 3 | 8wk intervals | N/A | CHMI | iv | 3.2 x 10³ PfSPZ (PfSPZ Challenge) | Homologous | 3 or 11wk | 5/5 | N/A | N/A | 10.1093/cid/ciz1152 |
| | | | | | 6 | | 1.8 x 10⁶ spz | | | | | | | | 7.5wk | 2/6 | N/A | N/A | |
| | 1 | NCT02627456 | Adults (M & F; 18-50yo) | Malaria-exposed (Mali) | 55 | iv | 1.8 x 10⁶ spz | 3 | 1, 13 and 19wk intervals | N/A | Natural transmission | N/A | N/A | N/A | 24wk surveillance post-vaccination | 23/55 | N/A | N/A | 10.1016/S1473-3099(21)00332-7 |
| | 2 | NCT02687373 | Infants (M & F; 5-12mo) | Malaria-exposed (western Kenya) | 67 | iv | 4.5 x 10⁵ spz | 3 | 8wk intervals | N/A | Natural transmission | N/A | N/A | N/A | 6mo after the last immunization (1st statistical end point) | 22/64 | N/A | N/A | 10.1038/s41591-021-01470-y |
| | | | | | 65 | | 9.0 x 10⁵ spz | | | | | | | | | 18/61 | N/A | N/A | |
| | | | | | 68 | | 1.8 x 10⁶ spz | | | | | | | | | 28/67 | N/A | N/A | |
| | 2 | NCT02601716 | Adults (M & F; 18-45yo) | Malaria-naïve | 15 | iv | 4.5 x 10⁵ spz | 5 | Days 1, 3, 5 and 7, and wk 16 (multi-dose priming and delayed boosting) | N/A | CHMI | Mosquito bite | 5 mosquitoes | Heterologous | 12wk after the last immunization | 6/15 | Not protected volunteers were boosted at 12wk prior to repeat CHMI at 24wk | 3/6 | 10.1093/cid/ciaa1294 |
| | | | | | 15 | | 9.0 x 10⁵ spz | 3 | 8wk intervals | | | | | | | 3/15 | | 6/8 | |
| | | | | | 13 | | 1.8 x 10⁶ spz | 3 | | | | | | | | 3/13 | N/A | N/A | |
| | | | | | 14 | | 2.7 x 10⁶ + 9.0 x 10⁵ spz | 1 + 2 | | | | | | | 24wk after the last immunization | 3/14 | N/A | N/A | |
| | | | | | 17 | | | 5 | Days 1, 3, 5, 7 and 113 | | | | | | | 7/17 | N/A | N/A | |

| Type of WSpz vaccine | Clinical trial phase | NCT | Age group | Malaria-naïve or -exposed subjects | Nr. of subjects | Administration route | Dose | Number of doses | Schedule | Antimalarial drug | Type of challenge | Administration route | Dose | Homologous/ Heterologous | Time-point | Nr. protected subjects/Total | Information | Nr. protected subjects/Total | DOI |
|---|---|---|---|---|---|---|---|---|---|---|---|---|---|---|---|---|---|---|---|
| | 1 | NCT03590340 | Adults (M & F; 18-32yo) | Malaria-exposed (Equatorial Guinea) | 21 | iv | $9.0 \times 10^5$ spz | 4 | Days 1, 3, 5 and 7 | N/A | CHMI | iv | $3.2 \times 10^3$ PfSPZ (PfSPZ Challenge) | Homologous | 6-7wk after the last immunization | 10/21 | N/A | N/A | 10.4269/ajtmh.21-0942 |
| | | | | | 18 | | | 5 | Days 1, 3, 5, 7 and 29 | | | | | | | 11/18 | N/A | N/A | |
| | | | | | 21 | | | 3 | Days 1, 8 and 29 | | | | | | | 7/21 | N/A | N/A | |
| | 1 | NCT02704533 | Adults (M & F; 18-45yo) | Malaria-naïve | 5 | iv | $9.0 \times 10^5$ spz | 3 | Days 1, 8 and 29 | N/A | CHMI | iv | $3.2 \times 10^3$ PfSPZ (PfSPZ Challenge) | Homologous | 3wk after the last immunization | 5/5 | N/A | N/A | 10.1038/s41541-022-00510-z |
| | | | | | 6 | | | | | | | | | Homologous | | 4/6 | All volunteers were heterologously rechallenged 9-10wk after the last immunization. | 5/6 | |
| | | | | | 6 | | | | | | | | | Heterologous | | 5/6 | All volunteers were heterologously rechallenged 9-10wk after the last immunization. | 5/6 | |
| | | | | | 6 | | $1.35 \times 10^6$ spz | 2 | Days 1 and 8 | | | | | Homologous | | 4/6 | N/A | N/A | |
| | | | | | 6 | | $2.7 \times 10^6$ spz | | Days 1 and 8 | | | | | Homologous | | 3/6 | N/A | N/A | |
| | 1 | NCT02663700 | Adults (M & F; 21-40yo) | Malaria-exposed (Burkina Faso) | 39 | iv | $2.7 \times 10^6$ spz | 3 | 8wk intervals | N/A | Natural transmission | N/A | N/A | N/A | 24wk surveillance post-vaccination | 25/39 | N/A | N/A | 10.1126/scitranslmed.abj3776 |
| | | | | | | | | | | | | | | | 76wk surveillance post-vaccination | 9/39 | N/A | N/A | |
| | 1 | NCT03420053 | Adults (HIV-; M & F; 18-45yo) | Malaria-exposed (Tanzania) | 4 | iv | $9.0 \times 10^5$ spz | 5 | Days 1, 3, 5, 7 and 29 | N/A | CHMI | iv | $3.2 \times 10^3$ PfSPZ (PfSPZ Challenge) | Homologous | 3wk after the last immunization | 4/5 | N/A | N/A | 10.1172/JCI169060 |
| | | | Adults (HIV+; M & F; 18-45yo) | | 5 | | | | | | | | | | | | 0/4 | N/A | N/A | |
| | 1 | NCT03510481 | Adults (M & F; 18-38yo) | Malaria-exposed (Mali) | 49 | iv | $9.0 \times 10^5$ spz | 4 | Wk 0, 1, 4 and 42 | N/A | Natural transmission | N/A | N/A | N/A | 3year surveillance post-vaccination | Not accessible in the preprint | N/A | N/A | 10.2139/ssrn.4769103 |
| | | | | | 42 | | $1.8 \times 10^6$ spz | | Wk 0, 8, 16 and 54 | | | | | | | | Not accessible in the preprint | N/A | N/A | |
| | 2 | NCT03989102 | Adults (M & F; 18-38yo) | Malaria-exposed (Mali) | 94 | iv | $9.0 \times 10^5$ spz | 3 | Wk 0, 1 and 4 | N/A | Natural transmission | N/A | N/A | N/A | 1year surveillance post-vaccination | 40/94 | N/A | N/A | 10.2139/ssrn.4769103 |
| | | | Adults (F; 18-38yo) | | 85 | | | | | | | | | | | 2year surveillance post-vaccination | 32/85 | N/A | N/A | |
| | | | Adults (M & F; 18-38yo) | | 100 | | $1.8 \times 10^6$ spz | | | | | | | | | 1year surveillance post-vaccination | 51/100 | N/A | N/A | |
| | | | Adults (F; 18-38yo) | | 92 | | | | | | | | | | | 2year surveillance post-vaccination | 36/92 | N/A | N/A | |
| **CPS** | colspan | | | | | | | | | **CPS immunization** | | | | | | | | | |
| | N/A | NCT00442377 | Adults (M & F; 18-45yo) | Malaria-naïve | 10 | Mosquito bite | 12-15 bites from Pf-infected mosquitoes | 3 | 1mo intervals | Chloroquine | CHMI | Mosquito bite | 5 mosquitoes | Homologous | 8wk after the last immunization 4wk after chloroquine discontinuation | 10/10 | 6 protected subjects underwent rechallenge 2.5 years after the last immunization | 4/6 | 10.1056/NEJMoa0805832  10.1016/S0140-6736(11)60360-7 |
| | N/A | NCT01236612 | Adults (M & F; 18-35yo) | Malaria-naïve | 5 | Mosquito bite | 15 bites from Pf-infected mosquitoes | 3 | 1mo intervals | Chloroquine | CHMI | Mosquito bite | 5 mosquitoes | Homologous | 21wk after the last immunization 17wk after chloroquine discontinuation | 5/5 | N/A | N/A | 10.1073/pnas.1220360110 |
| | | | | | 9 | | | | | | | iv | 1962 | Pf-infected erythrocytes | | 0/9 | | | |
| | N/A | NCT01422954 | Adults (M & F; 18-35yo) | Malaria-naïve | 5 | Mosquito bite | 8 bites from Pf-infected mosquitoes | 3 | 1mo intervals | Chloroquine | CHMI | Mosquito bite | 5 mosquitoes | Homologous | 20wk after the last immunization 16wk after prophylaxis discontinuation | 3/5 | N/A | N/A | 10.1371/journal.pone.0112910 |
| | | | | | 10 | | | | | Mefloquine | | | | | | 7/10 | | | |
| | N/A | NCT01660854 | Adults (M & F; 18-45yo) | Malaria-naïve | 5 | Mosquito bite | 15 bites from Pf-infected mosquitoes | 3 | 1mo intervals | Chloroquine | CHMI | Mosquito bite | 5 mosquitoes | Homologous | 19wk after the last immunization 15wk after chloroquine discontinuation | 4/5 | 13 subjects were heterologously rechallenged (Pf NF135.C10 strain) 14mo after the last immunization. | 1/2 | 10.1371/journal.pone.0124243 |
| | | | | | 9 | | 10 bites from Pf-infected mosquitoes | | | | | | | | | 8/9 | | 1/7 | |
| | | | | | 10 | | 5 bites from Pf-infected mosquitoes | | | | | | | | | 5/10 | | 0/4 | |
| | N/A | NCT02098590 | Adults (M & F; 18-35yo) | Malaria-naïve | 5 | Mosquito bite | 15 bites from Pf-infected mosquitoes | 3 | 1mo intervals | Chloroquine | CHMI | Mosquito bite | 5 mosquitoes | Homologous | 18wk after the last immunization 14wk after chloroquine discontinuation | 5/5 | N/A | N/A | 10.1186/s12916-017-0923-4 |
| | | | | | 10 | | | | | | | | | Heterologous (NF135.C10) | | 2/10 | | | |
| | | | | | 9 | | | | | | | | | Heterologous (NF166.C8) | | 1/9 | | | |
| | colspan | | | | | | | | | **Sanaria® PfSPZ-CVac** | | | | | | | | | |
| | 1/2b | NCT01728701 | Adults (M & F; 18-35yo) | Malaria-naïve | 10 | id | $7.5 \times 10^4$ spz | 3 | Days 8, 36 and 64 | Chloroquine | CHMI | Mosquito bite | 5 mosquitoes | Homologous | 60d after the last immunization 33d after chloroquine discontinuation | 2/10 | N/A | N/A | 10.4269/ajtmh.15-0621 |
| | | | | | 4 | | $7.5 \times 10^4$ spz | 4 | Days 8, 36, 64 and 232 | Chloroquine | | | | | 137d after the last immunization 109d after chloroquine discontinuation | 0/4 | N/A | N/A | |
| | 1 | NCT02115516 | Adults (M & F; 18-45yo) | Malaria-naïve | 9 | iv | $3.2 \times 10^3$ spz  $1.28 \times 10^4$ spz  $5.12 \times 10^4$ spz | 3 | 28d intervals | Chloroquine | CHMI | iv | $3.2 \times 10^3$ PfSPZ (PfSPZ Challenge) | Homologous | 8-10wk after the last immunization 5-7wk after chloroquine | 3/9  6/9  9/9 | N/A | N/A | 10.1038/nature21060 |
| | 1 | NCT02511054 NCT03083847 | Adults (M & F; 18-50yo) | Malaria-naïve | 6 | iv | $2 \times 10^5$ spz | 3 | 4wk intervals | Chloroquine | CHMI | iv | $3.2 \times 10^3$ PfSPZ (PfSPZ Challenge) | Heterologous | 13wk after the last immunization | 6/6 | N/A | N/A | 10.1038/s41586-021-03684-z |
| | | | | | 8 | | | | | Pyrimethamine | | | | Homologous | | 7/8 | | | |
| | | | | | 9 | | | | | | | | | Heterologous | | 7/9 | | | |
| | 1 | NCT02996695 | Adults (M & F; 18-45yo) | Malaria-exposed (Mali) | 29 | iv | $2.948 \times 10^5$ spz | 3 | 4wk intervals | Chloroquine | Natural transmission | N/A | N/A | N/A | 24wk surveillance post-vaccination | 16/29 | N/A | N/A | 10.1016/j.eclinm.2022.101579 |
| | 1 | NCT02859350 | Adults (M & F; 18-35yo) | Malaria-exposed (Equatorial Guinea) | 15 | iv | $2.7 \times 10^6$ spz PfSPZ Vaccine | 3 | 8wk intervals | N/A | CHMI | iv | $3.2 \times 10^3$ PfSPZ (PfSPZ Challenge) | Homologous | 15wk after the last immunization | 5/15 | N/A | N/A | 10.4269/ajtmh.20-0435 |
| | | | | | 13 | | $1.0 \times 10^5$ spz PfSPZ-CVac | | 4wk intervals | Chloroquine | | | | | 14wk after the last immunization | 8/13 | | | |
| | 1 | NCT02115516 | Adults (M & F; 18-45yo) | Malaria-naïve | 8 | iv | $5.12 \times 10^4$ spz | 3 | 5d intervals | Chloroquine | CHMI | iv | $3.2 \times 10^3$ PfSPZ (PfSPZ Challenge) | Homologous | 10wk after the last immunization | 5/8 | N/A | N/A | 10.1038/s41541-022-00473-1 |
| | | | | | 9 | | | | 14d intervals | | | | | | | 6/9 | | | |
| | 1 | NCT02773979 | Adults (M & F; 18-45yo) | Malaria-naïve | 7 | iv | $5.12 \times 10^4$ spz | 3 | 7d intervals | Chloroquine | CHMI | iv | $3.2 \times 10^3$ PfSPZ (PfSPZ Challenge) | Homologous | 10wk after the last immunization | 0/7 | N/A | N/A | 10.1371/journal.ppat.1009594 |
| | | | | | 8 | | $1.024 \times 10^5$ spz | | 5d intervals | | | | | | | 6/8 | | | |
| | 1 | 2018-004523-36 | Adults (M & F; 18-45yo) | Malaria-naïve | 13 | iv | $1.1 \times 10^5$ spz | 3 | Days 1, 6 and 29 | Chloroquine | CHMI | iv | $3.2 \times 10^3$ PfSPZ (PfSPZ Challenge) | Heterologous | 12wk after the last immunization | 10/13 | N/A | N/A | 10.1038/s41467-021-22740-w |
| | 1 | NCT02858817 | Adults (M & F; 18-45yo) | Malaria-naïve | 8 | iv | $5.12 \times 10^4$ spz | 3 | 4wk intervals | Atovaquone-proguanil | CHMI | iv | $3.2 \times 10^3$ PfSPZ (PfSPZ Challenge) | Heterologous | 10wk after the last immunization | 2/8 | N/A | N/A | 10.1101/2020.09.14.296152 |
| | | | | | 10 | | $1.0 \times 10^5$ spz | | | | | | | | | 2/10 | | | |
| | colspan | | | | | | | | | **Sanaria® PfSPZ-GA1** | | | | | | | | | |
| **GAP** | 1/2a | NCT03163121 | Adults (M & F; 18-34yo) | Malaria-naïve | 12 | iv | $4.5 \times 10^5$ spz PfSPZ-GA1 | 3 | 8wk intervals | N/A | CHMI | Mosquito bite | 5 mosquitoes | Homologous | 3wk after the last immunization | 1/12 | N/A | N/A | 10.1126/scitranslmed.aaz5629 |
| | | | | | 13 | | $9.0 \times 10^5$ spz PfSPZ Vaccine | | | | | | | | | 0/13 | | | |
| | | | | | 13 | | $9.0 \times 10^5$ spz PfSPZ-GA1 | | | | | | | | | 2/13 | | | |
| | colspan | | | | | | | | | **PfGAP3KO** | | | | | | | | | |
| | 2 | NCT03168854 | Adults (M & F; 18-50yo) | Malaria-naïve | 8 | Mosquito bite | ~200 mosquitoes | 5 | 4wk intervals and 8wk interval between doses 4 and 4 | N/A | CHMI | Mosquito bite | 5 mosquitoes | Homologous | 4wk after the last immunization | 4/8 | The 4 protected subjects underwent rechallenge ~26wk (6mo) after the 1st CHMI | 1/6 | 10.1126/scitranslmed.abn9709 |
| | | | | | 6 | | | 3 | 4wk intervals and 8wk interval between doses 2 and 3 | | | | | | | 3/6 | 2 protected subjects underwent rechallenge ~26wk (6mo) after the 1st CHMI | | |

M male, F female, yo years old, d days, wk weeks, mo months, avg average, N/A not applicable.
If not stated otherwise, the rechallenge was performed with a strain homologous to the vaccine strain.

# Achievements, challenges, and the path forward for WSpz malaria vaccination

Over the last ~15 years, the field of WSpz vaccination underwent significant progress, culminating in the generation of the PfSPZ family of regulatory-compliant injectable products. At the same time, progress in the genetic manipulation of malaria parasites has enabled the creation of transgenic sporozoites in which specific genes have either been abrogated to create multiple GAPs (Franke-Fayard et al, 2022; Goswami et al, 2020; Goswami et al, 2024; Kublin et al, 2017; Roestenberg et al, 2020b), or inserted to amplify their immunogenicity and scope of action (Mendes et al, 2018a; Miyazaki et al, 2020; Moita et al, 2022a). Concomitantly, methods to analyze the immune responses elicited in humans by WSpz vaccines have evolved (Moita et al, 2022b), enabling an increasingly thorough understanding of their underlying protective immunity.

Despite these major advances, WSpz vaccination still faces undeniable challenges, the answers to which will define the future of the field. We are now at a critical juncture, where these hurdles must be frontally addressed, with equivalent doses of optimism and realism. The limited availability of resources for malaria research (Prudencio and Costa, 2020) demands that the right choices be made, and that a clear path forward be defined.

## Alternative WSpz vaccine formulations: which should be prioritized?

Up to present, four types of Pf-based WSpz vaccines have been tested in the clinic, including early-arresting replication-deficient (EARD)-PfGAP, PfRAS, late-arresting replication-competent (LARC)-PfGAP, and PfCPS. Crucially, each of these have been, or will soon be, clinically assessed employing injectable formulations, PfSPZ-GA1, PfSPZ Vaccine, PfSPZ-LARC2 and PfSPZ-CVac, respectively. One important question going forward is therefore which of these formulations should be prioritized for future development. It has been suggested that the genetic homogeneity of GAP may confer PfGAP potential advantages over their PfRAS counterparts (Khan et al, 2012). On the other hand, PfCPS poses potential safety risks, resulting from the requirement for proper co-administration of a drug (Richie et al, 2023; Sahu et al, 2021). Ultimately, the prioritization of PfWSpz formulations must take into consideration not only these factors, but also pivotal issues like dose, protective efficacy, and durability of protection.

To try and address these issues, we recently carried out the first direct comparison of rodent Pb-based surrogates of the four PfWSpz vaccine types listed above, in a mouse model of immunization and infection. In these studies, the short- and long-term protection conferred by various doses of EARD-PbGAP *PbΔb9Δslarp* (van Schaijk et al, 2014), (ii) PbRAS (Jobe et al, 2009), LARC-PbGAP *PbΔmei2Δlisp2* (Moita et al, 2023b) and PbCPS, (Bijker et al, 2015) were assessed in parallel (Moita et al, 2023a; Moita et al, 2023b). The results obtained in these experiments indicated that EARD-PbGAP displays the lowest protective efficacy of the four formulations tested (Moita et al, 2023b), and that the most durable protection among these formulations is afforded by PbRAS and LARC-PbGAP (Moita et al, 2023a). Of note, the higher efficacy of LARC- compared to EARD-GAP has also been established in a human study in which the *PfΔmei2* LARC-PfGAP delivered by mosquito bite was shown to induce considerably greater protective immunity than its *PfΔb9Δslarp* EARD-PfGAP counterpart (Franke-Fayard, 2022), in agreement with the modest results of a trial employing an injectable formulation of the latter parasite line, termed PfSPZ-GA1 (Roestenberg et al, 2020a).

Although PfRAS remains the gold-standard of WSpz malaria vaccination, the conclusions from these studies and the general considerations made above support the prioritization of LARC-PfGAP for WSpz vaccination against malaria. In this context, the results of the ongoing clinical evaluation of PfSPZ-LARC2 Vaccine, a LARC-PfGAP created through the deletion of the *mei2* and *linup* genes from the *Pf* genome, as recently described in this journal (Goswami et al, 2024; Moita and Prudencio, 2024), is highly anticipated.

## Potency, administration regimen, and heterologous protection of PfSPZ Vaccine and PfSPZ-CVac

As noted above, both PfSPZ Vaccine and PfSPZ-CVac have been clinically tested at various doses and employing different administration regimens, with subjects being either homologous or heterologously challenged by controlled human malaria infection (CHMI) at varying times after the last immunization. Results from these trials revealed important differences in the vaccines' protective efficacy depending, or not, on those variables. For example, the 100% protection achieved by immunization with five doses of $1.35 \times 10^5$ PfSPZ Vaccine and homologous challenge 3 weeks after the last immunization (Seder et al, 2013) has not been attained in subsequent trials where higher vaccine doses and the same time-to-challenge were employed (Epstein et al, 2017; Mordmuller et al, 2022). Also of note, 100% heterologous protection has never been achieved for PfSPZ Vaccine. This vaccine has been employed in multiple immunization regimens, ranging from 2 to 5 doses of between $1.35 \times 10^5$ and $2.7 \times 10^6$ PfSPZ, yielding different protective efficacies. In general, protection induced by PfSPZ Vaccine appears to be dose-dependent up to doses of $9.0 \times 10^5$ immunizing parasites and seems to decrease thereafter (Jongo et al, 2020). The vast amounts of data generated by these trials has enhanced our ability to identify and select the most protective immunization regimens. It seems clear that efficacy is not merely dependent on the total number of sporozoites administered, but also on the number of, and the spacing between, administrations. Naturally, the practical advantages of condensed immunization regimens relative to those in which vaccination takes place over the course of several months must also be considered when selecting the most appropriate vaccination schemes going forward. In view of the accumulated results obtained in these trials, a three-dose condensed regimen of PfSPZ Vaccine administration on days 1, 8, and 29 appears to offer the best combination of efficacy and practicality, and has therefore been selected for further development (Richie et al, 2023).

Three monthly immunizations with $5.12 \times 10^4$ PfSPZ-CVac yielded 100% protection against homologous challenge (Mordmuller et al, 2017). This result shows that the dose of PfSPZ-CVac required for complete protection is markedly lower than that required for PfSPZ Vaccine, illustrating the superior potency of the former relative to the latter. It should be mentioned that, in a subsequent study, a similar vaccination regimen yielded 80 rather than 100% protection against homologous challenge (Mwakingwe-

Omari et al, 2021), highlighting the variability of the results obtained across different trials. However, and most importantly, three immunizations with $2 \times 10^5$ PfSPZ-CVac one month apart conferred 100% protection against heterologous challenge (Mwakingwe-Omari et al, 2021). This result not only constitutes the first (and, until now, the only) instance of complete protection against heterologous infection by any malaria vaccine candidate, but further substantiates the stronger protective efficacy of PfSPZ-CVac compared with even a 4.5-fold higher dose of PfSPZ Vaccine (Lyke et al, 2021). This highlights the fact that PfSPZ-CVac vaccination requires markedly lower numbers of immunizing parasites and has therefore a lower associated cost than the PfSPZ Vaccine. Another important aspect to bear in mind is that PfSPZ-CVac must not be administered when transient parasitemia from a previous dose is present, as this may completely abrogate the vaccine's protective efficacy (Murphy et al, 2021). Crucially, 75 and 77% protection against homologous and heterologous CHMI, respectively, has been achieved employing condensed regimens of administration of $1 \times 10^5$ PfSPZ-CVac (Mordmuller et al, 2022; Murphy et al, 2021), supporting the feasibility of a vaccination schedule that can be completed by inoculation of three vaccine doses within a 1-month period. Thus, such a condensed vaccination scheme enables simple immunization and drug regimens that include only three visits to the clinic, in which the antimalarial drug is administered just before each parasite injection (Richie et al, 2023).

Overall, these studies lend support to the use of condensed vaccination regimens for WSpz malaria vaccination, and suggest that PfSPZ-CVac's higher potency relative to PfSPZ Vaccine would allow for enhanced heterologous protection. However, as noted above, a vaccine that depends on the concomitant administration of an antimalarial drug poses safety risks that cannot be ignored. It is hoped that PfSPZ-LARC2 Vaccine might bring together the best of both worlds by displaying "the safety of PfSPZ Vaccine and the efficacy of PfSPZ-CVac" (Richie et al, 2023). Should that expectation be confirmed in the clinic, it would certainly represent a major achievement for the future of malaria vaccination.

## Protective efficacy in malaria-endemic regions, in children and in women of childbearing age

In its first clinical evaluation in a malaria-endemic region, the PfSPZ Vaccine was administered to adults in five doses of $2.7 \times 10^5$ immunizing parasites at days 0, 28, 56, 84, and 140. Vaccine efficacy against natural exposure in this trial was ~52% (Sissoko et al, 2017). This is similar to the protection observed in two subsequent trials in Africa employing three doses of $1.8 \times 10^6$ (Sissoko et al, 2022) or $2.7 \times 10^6$ (Sirima et al, 2022) PfSPZ Vaccine, but is markedly lower than the 80% protection observed in malaria-naive subjects to whom five doses of $2.7 \times 10^5$ PfSPZ Vaccine were also administered at days 0, 28, 56, 84, and 140 and heterologous CHMI was performed 3 weeks after the last immunization (Epstein et al, 2017). The differences in the protective efficacies of the PfSPZ Vaccine in malaria-naive vs malaria pre-exposed were made even more striking by a trial in Tanzanian adults in which the same vaccination regimen yielded only 20% protection against homologous CHMI three weeks after the last immunization (Jongo et al, 2018). Also highly relevantly, a recent study revealed five administrations of PfSPZ Vaccine afforded no protection to HIV + Tanzanian individuals, in contrast with 80% protective efficacy

observed for their HIV- counterparts undergoing the same vaccination regimen (Jongo et al, 2024).

A groundbreaking new study has demonstrated that the PfSPZ Vaccine provides durable protection against malaria in women of childbearing age, including during pregnancy, without the need for booster doses. In these trials conducted in Mali, three doses of $9.0 \times 10^5$ or $1.8 \times 10^6$ PfSPZ Vaccine administered at 4-week intervals, combined with presumptive malaria treatment before the first dose, demonstrated significant protection against Pf parasitemia and clinical malaria across two transmission seasons, while also protecting against pregnancy malaria. Relevantly, the vaccine was well tolerated by both mothers and their offspring, highlighting the potential of early immunization to substantially reduce maternal malaria infections and improve pregnancy outcomes (Diawara, et al., 2024).

A lower efficacy of equivalent WSpz vaccination regimens in pre-exposed vs naive individuals has also been observed for PfSPZ-CVac. Three doses of $1.0 \times 10^5$ PfSPZ-CVac administered at 4-week intervals yielded 55% protection against homologous CHMI in Equatoguinean adults (Jongo et al, 2020), in contrast with the 100% protection observed for three 4-week spaced administrations of $5.12 \times 10^4$ PfSPZ-CVac to malaria-naive volunteers (Mordmuller et al, 2017). Also, a subsequent trial in Mali revealed that three doses of $2 \times 10^5$ PfSPZ-CVac administered at the same intervals as above afforded only ~33% non-statistically significant protection against naturally transmitted Pf infection over 48 weeks (Coulibaly et al, 2022).

Equally of concern is the limited efficacy of the PfSPZ Vaccine in 5–12 months-old Kenyan infants immunized with three doses of $4.5 \times 10^5$, $9.0 \times 10^5$ or $1.8 \times 10^6$ of the vaccine spaced by 8 weeks. Although 41 and 46% protection against Pf infection and clinical malaria, respectively, was observed for the highest dose group at 3 months after the last immunization, no significant protection against infection was observed in any dose group at the 6 months primary endpoint (Oneko et al, 2021). These results suggest that immune responses to the PfSPZ Vaccine are age-dependent, which has been linked to a reduced potency of T-cell responses in infants (Simon et al, 2015).

Collectively, these results raise concerns about the efficacy of current PfSPZ Vaccine and PfSPZ-CVac vaccination regimens in malaria-endemic regions and, particularly, in children residing in those areas. While the exact reasons for the hyporesponsiveness of malaria pre-exposed relative to malaria-naive individuals to vaccination are not entirely clear, they have been linked to possible immune tolerance to *Plasmodium* antigens. In fact, antibody responses to PfCSP in semi-immune African adults were consistently and markedly lower than those observed in malaria-naive individuals immunized with the same PfSPZ regimens ((Sissoko et al, 2017; Sissoko et al, 2022) vs (Epstein et al, 2017; Lyke et al, 2021)), indicating that the latter respond significantly better to the vaccine than the former. Another factor that appears to strongly impact vaccine efficacy is the immunomodulatory effect of parasitemia at the time of immunization. Indeed, evidence that protection is highly dependent on parasite clearance prior to the first immunization is illustrated by the results of several clinical trials (e.g., (Sirima et al, 2022; Sissoko et al, 2017; Sissoko et al, 2022) vs (Coulibaly et al, 2022; Oneko et al, 2021)), and by a direct comparison of PfSPZ-CVac efficacy in the presence or absence of parasitemia (Murphy et al, 2021). Collectively, these observations

warrant the presumptive treatment of parasitemia before vaccination in ongoing and future trials in regions of malaria endemicity (Richie et al, 2023). Interestingly, however, this seems to be in apparent contradiction with a recent report suggesting that RTS,S displays significantly higher efficacy in parasitemic than in non-parasitemic patients at the time of the first vaccination (Arisue and Palacpac, 2024; Juraska et al, 2024).

## Vaccine production, deployment, and administration

For many years, WSpz vaccination against malaria was deemed unpractical, not least the dependence on mosquitoes as the production and delivery method for these vaccines, which was, understandably, considered technically impractical and logistically impossible to implement in malaria-endemic settings (Druilhe and Barnwell, 2007; Luke and Hoffman, 2003). This major impediment to further development of WSpz vaccines was overcome by the development of methods to produce cryopreserved, aseptic, purified Pf sporozoites suitable for parenteral injection into humans (Hoffman et al, 2010; James et al, 2022; Richie et al, 2023).

Although the advent of vialed formulations of WSpz vaccines was nothing short of an absolute game changer for the field, a few challenges remain. These include the need to ensure that production capabilities are compatible with demand, that a suitable liquid nitrogen vapor phase (LNVP) cold chain is adequately implemented; the availability of adequately trained staff to administer these vaccines intravenously, and, ultimately, that cost of goods (CoG) safeguard affordability for those in highest need of these vaccines. All these aspects were thoroughly and candidly discussed in a very recent review authored by several members of the International PfSPZ Consortium (Richie et al, 2023), to which the reader is referred for further details on strategies already implemented or currently being pursued to address each one of these challenges.

However, one particular aspect that is not addressed in detail in that review concerns the loss of sporozoite viability observed during cryopreservation. In fact, data from mouse models employing rodent *P. yoelii* parasites point to 7.4-fold loss in in vivo infectivity as a result of sporozoite cryopreservation (Roestenberg et al, 2013). Moreover, while a 25–30% difference in potency and viability of fresh vs cryopreserved Pf sporozoites has been estimated from in vitro studies (Roestenberg et al, 2013), 3.200 intravenously injected PfSPZ into human volunteers were required to achieve infections comparable to those resulting from the bites of five Pf-infected mosquitoes (Gomez-Perez et al, 2015), suggesting that cryopreservation leads to ~6.4-fold loss of Pf sporozoite infectivity (Ruben et al, 2013). Naturally, the relatively low proportion of sporozoites that remain viable leads to an increase in the total number of sporozoites required for vaccination, with obvious implications for CoG. Therefore, solving the viability issue of cryopreservation can have major impacts on the feasibility of scaling these operations to the point needed to make them viable beyond travelers and deployed military personnel. Importantly, it has recently been shown that improvements in cryopreservation efficiency may be possible (Bowers et al, 2022). Nevertheless, while such improvements might lead to a reduction in CoG, they might also entail the need for reappraisal of the vaccination regimens that have been optimized over the last few years, with potential impacts on licensure.

## Transgenic parasites for WSpz vaccination against *P. vivax*

By their very nature, WSpz vaccines depend on the availability of *Plasmodium* sporozoites, the mosquito salivary gland-resident forms of malaria parasites. While Pf sporozoites can be easily obtained under laboratory conditions, an in vitro system for long-term culture of blood stages Pv remains unavailable (Kumari and Sinha, 2023; Thomson-Luque et al, 2017). Thus, Pv sporozoites can only be obtained from mosquitoes fed on the blood of Pv-infected patients (Bermudez et al, 2018), severely limiting the development of Pv-based WSpz vaccines.

In this context, parasites of other *Plasmodium* species engineered to express Pv antigens emerge as attractive alternatives for pre-clinical and clinical development of PvWSpz vaccine surrogates. Miyazaki et al generated a Pf parasite line, termed *Pf-Pv*CSP, that expresses a chimeric version of PvCSP's two major alleles, VK210 and VK247 (Miyazaki et al, 2020). Immunization of mice with *Pf-Pv*CSP sporozoites induced antibodies against both PfCSP and PvCSP, suggesting that they could be used in WSpz vaccination approaches to induce cross-protective immune responses against both these human malaria parasites.

An alternative surrogate PvWSpz vaccine relies on the use of genetically modified Pb parasites, engineered to express Pv antigens. The use of transgenic Pb sporozoites as immunization agents for human malaria has been pre-clinically (Mendes et al, 2018a; Mendes et al, 2018b) and clinically (Reuling et al, 2020) validated through PbVac, a PfCSP-expressing Pb parasite line. More recently, a similar principle was applied to Pv through the generation of PbViVac, a Pb parasite that expresses the VK210 variant of PvCSP (Moita et al, 2022a). Immunization of mice with sporozoites of this line elicited the production of antibodies that efficiently recognize and bind to immobilized Pv sporozoites (Moita et al, 2022a). This, combined with the cellular immunity presumed to arise from the high degree of CD8+ T-cell epitope similarity that exists between Pb and Pv (Moita et al, 2022a), warrants the expectation that Pb-based WSpz vaccines may be functionally protective against Pv infection.

While the prospect of producing Pv sporozoites in high numbers appears somewhat distant, transgenic Pf or Pb parasites constitute promising alternatives to the development of WSpz vaccines targeting that parasite. While the former of these two options potentially offers the advantage of simultaneous protection against both Pf and Pv, it requires the adequate knock-out of genes to ensure parasite attenuation, and the concomitant knock-in of genes that express the desired Pv antigen(s). Conversely, the "naturally attenuated" nature of Pb-based WSpz vaccines makes them inherently safe, and the lower stringency of containment requirements for the production of Pb-infected mosquitoes compared to that of their Pf-infected counterparts is expected to impact CoG. Moreover, Pb's high amenability to genetic modification and the presence of several neutral loci in this parasite's genome offer ample options for the insertion of multiple Pv antigens, making this a most attractive option for further development. Thus, various transgenic Pb parasites expressing different Pv antigens have been designed, including one line that expresses a chimeric version of the VK210 and VK247 variants of PvCSP. The latter line is currently being employed in non-human primate immunization experiments

to ascertain their protective efficacy in this model, paving the way for its future clinical evaluation.

## Conclusion and final remarks

Once practically unimaginable, the prospect of licensure of WSpz vaccines now appears to be within sight. Given the magnitude of the burden imposed by malaria, the addition of such a tool to the antimalarial armamentarium would constitute a tremendous victory in the fight against this devastating disease. Plans to implement PfSPZ vaccines targeting several population groups, including travelers to malaria-endemic regions, women of child-bearing potential, pregnant women and children living in areas of malaria endemicity, have recently been announced (Richie et al, 2023).

Nevertheless, it seems clear that many of the challenges outlined above still need to be adequately addressed before this vision can become a reality. The lower protective efficacy in malaria pre-exposed individuals compared to malaria-naive volunteers, the absence of protection in HIV+ individuals, and the disappointing results of the first trial conducted in infants raises justified concerns about the licensure of these vaccines for these target populations. At present, the more realistic prospect seems to be that of licensure for adult travelers from malaria-free to malaria-endemic regions in the relatively near future.

In the meantime, planned and ongoing clinical studies will seek to tackle some of the issues encountered in previous Phase 2 trials, and overcome the limited vaccine efficacy observed in vaccinated individuals naturally exposed to the disease. The presumptive treatment of parasitemia prior to the first immunization of adults is likely to play an important role in this regard, but it remains to be seen whether this will also make a significant difference in infant populations with immature immune systems. Additionally, clear immune correlates of WSpz vaccine efficacy are yet to be identified and must continue to be sought. At the same time, efforts will proceed toward the development of transgenic parasites expressing selected antigens for WSpz vaccination against Pv. Last but not least, there is amply justified reason to expect that, pending the results of its ongoing clinical evaluation, PfSPZ-LARC2 may swiftly assume center stage in the field of WSpz malaria vaccination.

The road traveled by WSpz malaria vaccines so far has been a long and occasionally bumpy one. It has been a journey of resilience and perseverance, guided by the desire to combat a disease that has fittingly been called a "scourge of humanity" (Cowman and Duraisingh, 2001). Although this journey is far from over, looking back at how many hurdles have already been successfully overcome, one cannot help but feel that the reality of WSpz malaria vaccination is indeed possible and has never been closer.

### Pending issues

i. Can sporozoite cryopreservation efficiency be improved and, if so, what would be the implications for currently optimized vaccination regimens?

ii. Can presumptive treatment of parasitemia before WSpz vaccination deliver the desired protection in infants and children, or is the low efficacy observed in previous clinical trials inherent to this population's immature immune system?

iii. Can a reliable correlate (signature) of protection afforded by WSpz vaccines be identified?

iv. What is the ideal boosting frequency for WSpz vaccines in the field?

v. Can PfSPZ supply be enhanced by, for instance, in vitro production of sporozoites?

vi. Can vaccine efficacy be enhanced by adjuvants, either added to or encoded by the immunizing parasites?

vii. Can WSpz vaccine dose or adjuvanting be optimized to enable vaccination by intramuscular injection at an acceptable cost?

viii. Is vaccination with genetically modified Pb sporozoites expressing Pv antigens protective against Pv infection in humans?

## Peer review information

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

## Acknowledgements

MP acknowledges the "la Caixa" Foundation for Grant HR21-848 and the European Union Horizon Europe programme under grant agreement No. 101080744. DM acknowledges funding from Fundação para a Ciência e Tecnologia, Portugal (SFRH/BD/144817/2019).

## Author contributions

**Diana Moita**: Conceptualization; Resources; Visualization; Writing—original draft; Writing—review and editing. **Miguel Prudêncio**: Conceptualization; Supervision; Writing—original draft; Project administration; Writing—review and editing.

## Disclosure and competing interests statement

The authors declare no competing interests.

