## [Peer Review File · EMBO Molecular Medicine]

Whole-sporozoite malaria vaccines: where we are, where we are going

Diana Moita and Miguel Prudêncio

Corresponding author: Miguel Prudêncio (mprudencio@medicina.ulisboa.pt)

Review Timeline:

Submission Date:	12th May 24
Editorial Decision:	2nd Jul 24
Revision Received:	11th Jul 24
Editorial Decision:	6th Aug 24
Revision Received:	7th Aug 24
Accepted:	14th Aug 24

Editor: Poonam Bheda

Transaction Report:

2nd Jul 2024

Dear Dr. Prudêncio,

Thank you for your submission to EMBO Molecular Medicine.

Please find below the two sets of comments I have now received regarding your review. As you will see, the referees are positive about its timeliness and suitability for publication. However, they do raise several important concerns that we would like you to address in a revision. In particular both reviewers commented on the promotional statements of Sanaria. Please tone these down so as not to distract from the scientific details and indicate whether you have a conflict of interest/vested interest in Sanaria.

Please also ensure that all other reviewer concerns are addressed. We may send your Review back to the Reviewers to ensure accuracy of scientific statements and citations.

- 1) a .doc formatted version of the manuscript text (including Figure legends and tables)
- 2) Separate figure files

We will send the article to our graphic artist who will edit/re-draw the figures for style and clarity. Therefore, please ensure the information shown is scientifically accurate and upload the file as a PDF (or SVG, or EPS), PowerPoint or Keynote in which the labels and objects are still editable. For figures created using Adobe Illustrator, please send the Illustrator (.ai) file. You can also send these to me by email (or share via link for files that are too big) so that we can already send these to the graphic designer to prevent delay in publishing your manuscript.

- 3) a letter INCLUDING the reviewer's reports and your detailed responses to their comments.
- 4) Glossary: EMBO Molecular Medicine articles will be accompanied by a glossary explaining some of the terms used for laymen. Could you please help us in identifying terms that may need an "explanation" that can be added to the glossary.
- 5) For more information: This is a short list of related web links for further consultation by the readers. Could you identify some relevant ones? Examples are patient associations, OMIM related links, databases, authors websites, etc.
- 6) Pending issues: At the end of each article we will have a box highlighting issues that still need further studies and where research efforts should converge (we call this the Pending issues box).
- 7) Disclosure and competing interest statement: Please include a statement declaring any competing commercial interests in relation to your submitted work. We updated our journal's competing interests policy in January 2022 and request authors to consider both actual and perceived competing interests. Please review the policy <https://www.embopress.org/competing-interests> and update your competing interests if necessary.
- 8) Up to 5 keywords

If you have any questions, please don't hesitate to ask.

Yours sincerely,

Poonam Bheda

Poonam Bheda, PhD
Scientific Editor
EMBO Molecular Medicine

***** Reviewer's comments *****

Referee #1 (Remarks for Author):

The review article by D Moita and M Prudencio, "Whole-sporozoite malaria vaccines: where we are, where we are going," describes the current development of a leading malaria vaccine strategy: the whole-sporozoite malaria vaccine. This review is well written and covers the development of these vaccines since they were described for the first time and the current clinical trials. The website created by the authors is helpful to the community and allows an overview of the current state of the art. Nevertheless, I have some significant issues that should be resolved before publication:

1. The allusion to Sanaria, Inc.'s role is interpellant throughout the review and should be toned down. Do the authors have a conflict of interest there? In that case, it should be stated.
2. It would be expected for a review to be more critical with the results that have resulted from clinical trials for these approaches. For example:
 - The parasite doses that are used to immunize volunteers are high. How a dose of 10^5 is different from one of 10^6 ? Is the immunization better when a higher number of parasites is used? For example, what are the scientific reasons by which one dose is determined as the optimal one in ref 32?
 - Different clinical trials have resulted in very heterogeneous results, not only in healthy volunteers but also in endemically-based individuals. The authors do not discuss the different plausible reasons for this in detail, and they should address them. Is it the genetic background or the presence of a specific immunity against malaria?
3. The chapter dedicated to WSpz vaccination against vivax is small and should be extended to explain the problem and the relevance of immunization with Pb murine parasites.

Finally, I would like to stress that it appears clear that the authors have some conflict of interest in the description of the current bibliography. The review can be published, but only after an in-depth review.

Referee #2 (Remarks for Author):

Moita and Prudencio offer a terrific summary and resource describing efforts to develop whole-sporozoite malaria vaccines. The manuscript is clearly and nicely written, and the comments and suggestions below are offered to provide a few additional opportunities to make the piece more approachable to non-experts. In addition to these content-based suggestions, I would also encourage tempering the language describing Sanaria to be less enthusiastically promotional, as it could detract from the important scientific details described here.

Page 2 - While some reports indicate that R21/Matrix M is more effective than RTS,S, there are important differences that prevented them from being evaluated in a direct, head-to-head manner. Consider if statements such as "presumably more efficacious" is warranted.

Page 2 - In describing the potential impacts of deploying partially effective vaccines, consider the work of Andrew Read and colleagues (PMID: 26214839) that provide a model that overall pathogen virulence can increase when incompletely effective vaccines are deployed.

Page 3 - It would be useful to list out the specific hurdles to whole pathogen vaccine approaches instead of talking about them categorically, and then with focuses on a few of them. For instance, significant description is offered on the problems due to cryopreservation, but more could be said about what the other hurdles are, and what their overall impact is.

Page 3 - In describing why whole pathogen vaccine strategies diminished in the 1970s and 1980s, consider that another major driver of this was the growing success, popularity, and ease-of-production of subunit vaccine approaches.

Pages 7-8 - Excellent introduction of the potential shortcomings of current cryopreservation approaches. It would be worth extending the line of reasoning and its impact to its logical conclusions: If the proportion of sporozoites that remain viable is low, the total number of sporozoites required for administration to each patient will be much higher, thus increasing production requirements and associated costs. Therefore, solving the viability issue of cryopreservation can have major impacts on the feasibility of scaling these operations to the point needed to make them viable beyond travelers and deployed military personnel.

Referee #1 (Remarks for Author):

The review article by D Moita and M Prudencio, "Whole-sporozoite malaria vaccines: where we are, where we are going," describes the current development of a leading malaria vaccine strategy: the whole-sporozoite malaria vaccine. This review is well written and covers the development of these vaccines since they were described for the first time and the current clinical trials. The website created by the authors is helpful to the community and allows an overview of the current state of the art.

We thank the reviewer for their positive remarks about the manuscript and appreciate the suggestions for its further improvement.

Nevertheless, I have some significant issues that should be resolved before publication:

1. The allusion to Sanaria, Inc.'s role is interpellant throughout the review and should be toned down. Do the authors have a conflict of interest there? In that case, it should be stated.

We note the reviewer's concern, and, in the interest of full transparency, we would like to clarify that an ongoing collaboration between the authors' laboratory and Sanaria, Inc. does exist, but that this does not constitute any conflict of interest. The various references to Sanaria, Inc. throughout the text are a direct result of this company's extensive contributions to whole-sporozoite malaria vaccination. In fact, as the sole entity worldwide that possesses the technology to produce GMP-compliant sporozoites suitable for injection in humans, they are an inescapable player in this field. Nevertheless, the authors do acknowledge that the 11 allusions to Sanaria, Inc. throughout were indeed excessive, and we have therefore modified the text to tone them down, with the company's name now being explicitly mentioned only twice throughout the manuscript.

2. It would expected for a review to be more critical with the results that have resulted from clinical trials for these approaches. For example:

-The parasite doses that are used to immunize volunteers are high. How a dose of 10^5 is different from one of 10^6 ? Is the immunization better when a higher number of parasites is used? For example, what are the scientific reasons by which one dose is determined as the optimal one in ref 32?

We thank the reviewer for this remark, and we have now made several additions to the text to address the issues pointed out.

Different clinical trials have resulted in very heterogeneous results, not only in

healthy volunteers but also in endemically-based individuals. The authors do not discuss the different plausible reasons for this in detail, and they should address them. Is it the genetic background or the presence of a specific immunity against malaria?

We are grateful for the reviewer's suggestion and have now modified the text to provide a more detailed appraisal of the differential protective efficacies observed in malaria-naïve and -pre-exposed individuals.

3. The chapter dedicated to WSpz vaccination against vivax is small and should be extended to explain the problem and the relevance of immunization with Pb murine parasites.

We thank the reviewer for this remark, and we respectfully note that prior work on the use of Pb parasites as surrogates from WSpz vaccination against Pv, and the possible advantages of this approach, were already described in the text. Nevertheless, following up on the reviewer's suggestion, have now added further information about this matter to the text.

Finally, I would like to stress that it appears clear that the authors have some conflict of interest in the description of the current bibliography. The review can be published, but only after an in-depth review.

We refer to our answer to the reviewer's first question, above, and would like to reinforce the absence of a conflict of interest with regards to Sanaria, Inc.

Referee #2 (Remarks for Author):

Moita and Prudencio offer a terrific summary and resource describing efforts to develop whole-sporozoite malaria vaccines. The manuscript is clearly and nicely written, and the comments and suggestions below are offered to provide a few additional opportunities to make the piece more approachable to non-experts.

We are grateful for the reviewer's very positive remarks about the manuscript, as well as for their constructive remarks and suggestions.

In addition to these content-based suggestions, I would also encourage tempering the language describing Sanaria to be less enthusiastically promotional, as it could detract from the important scientific details described here.

We kindly refer the reviewer to our answer to reviewer 1's first question, and confirm that references to Sanaria, Inc. have been substantially toned down.

Page 2 - While some reports indicate that R21/Matrix M is more effective than RTS,S, there are important differences that prevented them from being evaluated in a direct, head-to-head manner. Consider if statements such as "presumably more efficacious" is warranted.

We thank the reviewer for this suggestion, and have modified the manuscript text accordingly.

Page 2 - In describing the potential impacts of deploying partially effective vaccines, consider the work of Andrew Read and colleagues (PMID: 26214839) that provide a model that overall pathogen virulence can increase when incompletely effective vaccines are deployed.

We thank the reviewer for this suggestion and have now added a reference to Read et al.'s work to the manuscript.

Page 3 - It would be useful to list out the specific hurdles to whole pathogen vaccine approaches instead of talking about them categorically, and then with focuses on a few of them. For instance, significant description is offered on the problems due to cryopreservation, but more could be said about what the other hurdles are, and what their overall impact is.

Page 3 - In describing why whole pathogen vaccine strategies diminished in the 1970s and 1980s, consider that another major driver of this was the growing success, popularity, and ease-of-production of subunit vaccine approaches.

We thank the reviewer for both these suggestions, and have now added the either of these pieces of information to the text.

Pages 7-8 - Excellent introduction of the potential shortcomings of current cryopreservation approaches. It would be worth extending the line of reasoning and its impact to its logical conclusions: If the proportion of sporozoites that remain viable is low, the total number of sporozoites required for administration to each patient will be much higher, thus increasing production requirements and associated costs. Therefore, solving the viability issue of cryopreservation can have major impacts on the feasibility of scaling these operations to the point needed to make them viable beyond travelers and deployed military personnel.

We are grateful for the reviewer's remark and have now modified the text in accordance with their suggestion.

6th Aug 2024

Dear Dr. Prudêncio,

Thank you for the submission of your revised review to EMBO Molecular Medicine. Your manuscript has now been re-reviewed by the two original reviewers. Based on their advice (included below), I am pleased to inform you that we will be able to accept your manuscript pending the following final amendments and appropriate response to reviewers:

- 1) Subject categories: These can be removed from the review as we will include subject categories in the submission system.
- 2) Author contributions: Please remove from the manuscript and specify author contributions in our submission system. CRediT has replaced the traditional author contributions section because it offers a systematic machine-readable author contributions format that allows for more effective research assessment. You are encouraged to use the free text boxes beneath each contributing author's name to add specific details on the author's contribution. More information is available in our guide to authors:
<https://www.embopress.org/page/journal/17574684/authorguide#authorshipguidelines>
- 3) References: Please correct the reference citation in the reference list. Where there are more than 10 authors on a paper, note that only 10 will be listed, followed by "et al.". Please check "Author Guidelines" for more information.
<https://www.embopress.org/page/journal/17574684/authorguide#referencesformat>
- 4) Please rename "Conflict of Interest" to "Disclosure and competing interests statement". In addition, to address Reviewer 1's concerns on a conflict of interest with Sanaria/mentioning Sanaria in the text, we would suggest that you either keep the 2 mentions of Sanaria in the text and state that you collaborate with Sanaria in the Disclosure and competing interests statement or you remove the 2 mentions of Sanaria and simply cite the work (in which case you would not need to mention your collaboration with Sanaria).
- 5) Please include a letter INCLUDING my comments as well as the reviewer's reports and your detailed responses.

If you have any questions, please don't hesitate to ask.

Yours sincerely,

Poonam Bheda

Poonam Bheda, PhD
Scientific Editor
EMBO Molecular Medicine

***** Reviewer's comments *****

Referee #1 (Remarks for Author):

The revised article by D Moita and M Prudencio, "Whole-sporozoite malaria vaccines: where we are going," is improved, and the authors implemented the changes from two reviewers. I still have a significant problem with Sanaria citations:

I suggest removing citations to Sanaria from the whole text. I strongly recommend citing the papers that result from Sanaria technology/research, even if performed by Sanaria or their technology.

-Also, the authors mention that they have collaborated with Sanaria but don't acknowledge a conflict of interest; they should change this.

After these modifications, I think that the review can be published.

Referee #2 (Remarks for Author):

Thank you for being responsive to previous comments, and for adjusting your manuscript accordingly. This will be a nice addition to the literature on this central topic to our field.

Referee #1 (Remarks for Author):

The revised article by D Moita and M Prodencio, "Whole-sporozoite malaria vaccines: where we are going," is improved, and the authors implemented the changes from two reviewers. I still have a significant problem with Sanaria citations:

I suggest removing citations to Sanaria from the whole text. I strongly recommend citing the papers that result from Sanaria technology/research, even if performed by Sanaria or their technology.

As requested by the reviewer, both instances where Sanaria Inc. was mentioned in the text have been removed.

-Also, the authors mention that they have collaborated with Sanaria but don't acknowledge a conflict of interest; they should change this.

After these modifications, I think that the review can be published.

Since no references to Sanaria, Inc. remain in the text, we declare no Conflicts of Interests in the manuscript.

Referee #2 (Remarks for Author):

Thank you for being responsive to previous comments, and for adjusting your manuscript accordingly. This will be a nice addition to the literature on this central topic to our field.

We are grateful for the reviewer's nice remarks and overall opinion about the manuscript.

14th Aug 2024

Dear Dr. Prudêncio,

We are pleased to inform you that your manuscript is accepted for publication and is now being sent to our publisher to be included in the next available issue of EMBO Molecular Medicine.

Your manuscript will be processed for publication by EMBO Press. It will be copy edited and you will receive page proofs prior to publication.

You will soon be contacted by Springer Nature to sign your publishing license. When you login to the customer service website, please use the following token to waive the article publication charges. Should you experience any difficulty, please email publishing@embo.org.

XXXXXXXXXXXXXX

Yours sincerely,

Poonam Bheda, PhD
Scientific Editor
EMBO Molecular Medicine